# Design Optimization of PCB-Based Rotary-Inductive Position Sensors

**DOI:** 10.3390/s22134683

**Published:** 2022-06-21

**Authors:** Aldi Hoxha, Mauro Passarotto, Gentjan Qama, Ruben Specogna

**Affiliations:** 1EMCLab, Polytechnic Department of Engineering and Architecture, University of Udine, 33100 Udine, Italy; hoxha.aldi@spes.uniud.it (A.H.); mauro.passarotto@uniud.it (M.P.); 2Renesas Electronics Europe GmbH, 85609 München, Germany; gentjan.qama.jc@renesas.com

**Keywords:** inductive position sensors (IPS), absolute encoders, digital resolvers, eddy currents, non-linear least-squares, sensor optimization, surface integral method

## Abstract

This paper introduces a novel methodology to optimize the design of a ratiometric rotary inductive position sensor (IPS) fabricated in printed circuit board (PCB) technology. The optimization aims at reducing the linearity error of the sensor and amplitude mismatch between the voltages on the two receiving (RX) coils. Distinct from other optimization techniques proposed in the literature, the sensor footprint and the target geometry are considered as a non-modifiable input. This is motivated by the fact that, for sensor replacement purposes, the target has to fit a predefined space. For this reason, the original optimization technique proposed in this paper modifies the shape of the RX coils to reproduce theoretical coil voltages as much as possible. The optimized RX shape was obtained by means of a non-linear least-square solver, whereas the electromagnetic simulation of the sensor is performed with an original surface integral method, which are orders of magnitude faster than commercial software based on finite elements. Comparisons between simulations and measurements performed on different prototypes of an absolute rotary sensor show the effectiveness of the optimization tool. The optimized sensors exhibit a linearity error below 0.1% of the full scale (FS) without any signal calibration or post-processing manipulation.

## 1. Introduction

The correct measurement of the angles is of crucial interest in many industrial, automotive, and aerospace applications in which transducers responses are used to implement a proper control procedure. Depending on the physics involved in the sensing mechanism, a categorization of these transducers is possible; we can, thus, recognize four main types of angle sensors: optical, capacitive, magnetic, and inductive sensors. Each has pros and cons, and the choice of one type rather than another depends on the application and the performance of the transducer.

Optical encoders are able to perform precise and high resolution position measurements [1]. Indeed, displacement measurements by means of optical interferometer can reach sub-nanometer or picometer accuracy [2]. On the contrary, they depend on the working environment as the presence of vibrations or dust may induce an incorrect output [3]. Typically, due to their complex design, these sensors are more expensive if compared to the previously mentioned types [4].

On the other hand, capacitive displacement sensors, such as capacitive rotary encoders, are cheaper and they exhibit a simpler design with respect to optical ones. More than the fact that they have a low power consumption, they also provide high precision measurements [5,6,7,8]. However, their capacitance depends on temperature and humidity, and if used in harsh environment where dust and moisture are also present, the measurements are unavoidably affected by these factors [9,10].

By exploiting the fabrication techniques of complementary metal oxide semiconductor (CMOS) technologies, Hall effect and magnetoresistive (MR) position sensors are widely used thanks to their small size, low cost, and the full integration with the front-end system. If coupled with a permanent magnet, the displacements of the MR or Hall sensor with respect to the magnet result in a variation of their electrical properties. Yet, these sensors are sensitive to external magnetic fields; moreover, as far as the Hall sensor is concerned, in which the magnetic field is provided by a non-uniformly distributed permanent magnet, calibration is mandatory [4,11,12,13].

Finally, inductive position sensors (IPS) exploit the variation of self or mutual inductance in order to determine the displacement of the object to be detected. The variable magnetic field generated by a transmitting (TX) coil is perturbed by a moving object called *target*. A set of receiving (RX) coils placed in the neighborhood of the transmitter will experience induced voltages that depend on the target position. A signal conditioning circuit acquires these signals in order to determine the position of the target.

The induced voltages in the RX coils depend mainly on the physical property of the target. Different types of targets can be used: target made by ferromagnetic materials, by LC resonators, or by conductive but not magnetic materials (i.e., ironless) [14,15]. Having a high magnetic permeability, ferromagnetic targets increase mutual coupling with RX coils, providing higher output signals. Unfortunately, they are also sensitive to static or slowly varying external magnetic fields [16,17]. Similarly, LC resonators can provide greater output signal values compared to ironless targets thanks to the higher Q factor. As a drawback, their sensitivity to temperature can lead to continuous shifts in resonant operating frequencies, thus yielding a noisy measurement. Last but not least, sensors based on ironless targets exploit eddy currents induced on a conductive target by the variable magnetic field produced by a transmitting coil. Typically, they provide weaker output signals when compared to the other sensor types mentioned above, but they are immune to stray external fields. This feature makes them very attractive for automotive and industrial applications [18].

Among different IPS types there are some characterizing features that are worth mentioning. Firstly, depending on the post-processing algorithm, the measurement may be *ratiometric* or *differential*. Secondly, depending on the geometry of the RX coils, in absence of a target, the induced voltage may be zero or different from zero [14]. Typically, the Linear Variable Differential Transformers (LVDT) or Rotary Variable Differential Transformer (RVDT) sensors perform a differential measurement, although ratiometric ones are preferred in order to solve problems related to thermal variations [19].

As far as this study is concerned, the focus is on an absolute ironless rotary IPS, which performs ratiometric measurements. Moreover, the sensor is designed with printed circuit board (PCB) technology. A pioneering work in this regard can be found in [20]. Indeed, PCBs have a low cost and a reduced size with respect to typical resolvers [21]. Being robust, contactless, and immune to thermal variations, it is particularly adapted for applications in which environmental conditions are harsh. Yet, when compared to optical and capacitive transducers, this type of sensor presents higher linearity errors. In order to reduce linearity errors, in this work, an original optimization procedure that optimizes RX coil geometry is proposed. Preliminary results in this regard can be found in [22]. Previously, in [23,24], the main focus has been the optimization of the target geometry that, however, is usually fixed in industrial applications given that the target has to fit some predefined space. An enabling technology for sensor optimization in a reasonable amount of time constitutes fast virtual prototyping of the sensor. By exploiting a fast and efficient simulation software based on the surface integral method, combined with a non-linear least-square solver, linearity errors lower than 0.1%FS are achieved. Preliminary results about the simulation method can be found in [25]. We emphasize that this linearity error is obtained without applying any calibration or post-processing manipulation such as offset compensation or induced voltage normalization. The important implication is that the front end circuitry that interfaces with the sensor can be very simple or avoided.

The paper is organized as follows. Section 2 describes the working principle of the sensor, providing an overview on the non-ideality effects that may occur during its operation. Section 3 focuses on the surface integral method used to simulate the sensor, which is hundreds of times faster than finite element commercial software for the same error tolerance. Section 4 describes different optimization approaches tested by the authors in order to identify the most efficient one in terms of linearity performance and computation time. Section 5 provides a comparison between the simulation and the measurements of different designs, not only showing the effectiveness of the simulation but also the difficulty in performing a perfect measurement due to non-ideality effects. Finally, in Section 6, the conclusions are drawn.

## 2. Working Principle of the Ratiometric Rotary Inductive Position Sensor

Let us consider the sensor in the absence of a metallic target, as depicted in Figure 1a. It consists of a transmitter (TX) and two receivers (RX). The TX coil is driven by a known alternate current $I_{s}$, thus generating a variable magnetic field. As the Faraday–Newmann law predicts, RX coils exhibit induced voltages when operating in open circuit. Twisted in sine and cosine shapes, the induced voltage on each receiver is zero. Indeed, by considering only one receiver, $RX_{sin}$ for instance, the magnetic field generated by the transmitter acts on two identical areas of the coil that, however, have opposite orientations. Figure 1a illustrates also the numerical sequence of the current path if $RX_{sin}$ was short circuited.

If, instead, a conductive target is positioned above the sensor, eddy currents are generated inside it. These eddy currents create a magnetic field that opposes the magnetic field produced by the TX coil. As a result, a shielding effect affects the portion of the domain of the sensor covered by the target. Thus, voltages induced on the RX coils depend on the target’s position. For instance, for the target position depicted in Figure 1b, the induced voltage on receiver $RX_{sin}$ reaches the maximum value, whereas by rotating the target by 90 degrees, the induced voltage in that receiver becomes zero. This results in a cosine response that depends on target position θ. The same working principle characterizes receiver $RX_{cos}$, but it is rotated in quadrature with respect to $RX_{sin}$, and the induced voltage of $RX_{cos}$ provides a sine spatial variation.

The position of the target can be determined with the following formula:(1)$$\theta^{meas}=atan\left(\frac{U_{rxsin}\left(\theta \right)}{U_{rxcos}\left(\theta \right)}\right),$$
in which $\theta^{meas}$ is the reconstructed angle, and $U_{rxsin}\left(\theta \right)$ and $U_{rxcos}\left(\theta \right)$ are the sine and cosine responses of the induced voltages on the first and the second receiver, respectively.

### Non-Ideality Effects

The description of the sensor working principle drawn before represents an ideal case; yet, actually, non-ideality effects such as non-linearity, skin effect, tilt of the target, presence of other conductors near the sensing region, and a non-uniform distribution of the magnetic field influence the sensor’s measurements. In what follows, a brief description and a strategy to reduce these effects are provided.

The linearity error is of crucial importance since it defines the variation of the response of the sensor from the ideal curve. The ideal case is the one wherein the response of the sensor depends linearly on the position of the target as predicted in (Equation 1). As far as the IPS sensors are concerned, the linearity error basically depends on the difference between the ideal induced voltage and the actual signal picked up by RXs. This behavior is very difficult to predict, because many geometric factors have a role in this respect, such as the lack of symmetry in the transmitter coil, the non-uniform distribution of the induced magnetic field, the target, and the fact that different portions of the RX coils are placed in different PCB layers connected by means of vias.

Second, by possessing finite conductivity, the shielding capability of a real target is reduced. This factor is expressed in terms of penetration or skin depth, a parameter that depends on frequency. Table 1 shows penetration depths for different conductors in a frequency band that ranges from 2.2 MHz to 5.6 MHz, which are values that are typically used for automotive applications. The skin depth is provided by the following:(2)$$\delta =\sqrt{\frac{1}{\pi f\sigma \mu }},$$
where σ is the target conductivity, μ the magnetic permeability, and *f* the working frequency of the sensor. If the thickness of the target is relatively small when compared to the penetration depth, the eddy currents generated on the target cannot completely oppose the external magnetic field that is only partially shielded by the target. Hence, the target thickness has to be at least greater than the penetration depth at the lowest operation frequency. For our purposes, a copper target with a thickness of 35 μm is used, which provides good shielding efficiency.

Furthermore, the measure is sensitive to the air gap between the sensor and the target because of the variations on mutual coupling. In fact, many proximity sensors exploit this principle in order to measure the distance from a transmitting coil to a target [26]. However, for our purposes, where the focus is on angle measurements, this distance must be fixed. If the target is not parallel to the sensor’s surface, then the reaction magnetic field generated by the eddy currents on the target will not uniformly oppose the magnetic field generated by the TX coil. This implies that the induced voltages on the receivers will present different amplitudes depending on the tilt during the revolution of the target.

Another aspect regarding the non-idealities is the possible presence of metallic parts in the neighborhood of the operating region of the sensor. This can be considered as an additional source of eddy currents, which is different from the target, that might perturb the measurements, thus increasing linearity errors [14]. Nevertheless, in most applications, the environment is controlled and compensation techniques or simulations, for instance with the method proposed in [27], may be performed in order to take into account or predict this non-ideality effect.

Figure 1a,b show the presence of two dummy exits as far as the receivers are concerned. They have the important role of making the entire sensor symmetric. Given that the output of the receiver concatenates a certain amount of flux, an *offset* is always present on the received signals. The dummy exits reduce this effect since they contribute with the same amount of concatenated flux as the real output of the receivers.

Finally, RX coils traces are split between the top and the bottom of the PCB and the electrical connection is maintained through vias. On the other hand, the target is present only on one side of the sensor, thus making the shielding effect different from one side to the other of the PCB. The optimization of the receiver coils, which will be described in Section 4, takes implicitly into account all these effects and it tries to correct the RX coil’s shape in order to compensate and reduce them.

## 3. Surface Integral Method

In order to correctly model the behaviour of an IPS, eddy currents induced in the metallic target have to be accurately computed. To achieve this goal, the *surface integral method* in [28] was chosen. Preliminary results with respect to the simulation method applied on inductive position sensors can be found in [25]. This formulation presents three main advantages with respect to the standard finite element (FE) codes:Since the conducting target thickness is much smaller than penetration depth δ, the induced current density can be considered as uniform across the layer’s thickness. It is, therefore, convenient to represent the thin conductor as a surface in which the eddy currents are tangent to it;It requires the discretization (i.e., meshing) of the metallic target only, avoiding the representation of the air surrounding the device to be studied. This fact yields a sensible speed-up both when building the model and during computations too;Since there is no mesh in insulators, there is no need to re-mesh the domain during the motion of the target. This provides speed-up at different orders of magnitude with respect to software based on FE.

To evaluate the performance of a design, a complete simulation consists of several repeated solutions of an eddy current problem for different target positions. Once the current density in the target has been computed for a given position, the overall magnetic vector potential A generated by the TX coil and by the target is calculated in correspondence of the RX coil’s path. The induced voltage on RX coils is evaluated as the integral of the magnetic vector potential on the centerline of the RX traces. This procedure is repeated for each *p* position of the target above the sensor and the envelope of the induced voltages on the two RX coils are collected in vectors $U_{\mathrm{rxsin}}\in R^{p}$ and $U_{\mathrm{rxcos}}\in R^{p}$, which are given to the optimizer. As an example, less than 40 s is sufficient for the simulation of 101 positions of the target above the sensor. On the contrary, 24 h is needed for the simulation of the same number of positions with a leading commercial software based on FE.

## 4. Novel Sensor Optimization

Theoretically, the induced voltages should have a sine and cosine response with the same amplitude during the revolution of the target. Unfortunately, this is not the case in a real device since the non-ideality effects, described in the Section 2, come out. The deviation from the ideal voltages leads to linearity errors that are higher than other types of sensors.

Attempts in improving the linearity error can be found in [23,24], where a rotary IPS has been simulated with the FE Method, and optimization algorithms, such as response surface method (RSM) and particle swarm optimization (PSO), are used for searching for the optimal geometrical parameter of the device that minimize the linearity error defined as follows:(3)$$\epsilon_{\%}=100\cdot \frac{{∥\theta^{\mathrm{meas}}-\theta^{\mathrm{id}}∥}_{\infty }}{\theta_{FS}},$$
where $\theta^{\mathrm{meas}}$ and $\theta^{\mathrm{id}}$ are, respectively, the measured and ideal angles for each position of the target.

In [23], the RSM is used to extract an approximated model of the linearity where the model parameters are those that mostly affect the linearity error. The model is obtained by means of a set of 20 experiments and approximated with a second order polynomial where the independent variables are the radian, the thickness of the rotor, and the gap of the rotor from the stator. However, no optimization is performed on the stator side.

Unlike the RSM algorithm, PSO has the advantage of providing a global optimal solution. To identify the relevant parameters that mainly affect the sensor’s linearity, in [24], a screening procedure was performed that consists of a complete simulated target sweep for each geometrical parameter under investigation, such as excitation coil turn number, coil width, rotor thickness, etc. After the screening phase, only a subset of the parameters to be optimized was selected.

Both these methodologies are time consuming since the screening phase is performed with FEM. Moreover, since only the rotor parameters are optimized, the geometry of the sensor is less flexible. If any geometric parameter changes, for instance, the stator dimension, the screening procedure has to be repeated.

In contrast, in this paper, the geometry of the rotor was considered as a fixed input. This is something that occurs in most practical cases, where the target has to fit some predefined space or the gap for some reasons is set to a specific value. In this setting, the optimization techniques proposed thus far cannot be used. The main novelty of this paper is to propose a new methodology for sensor optimization that can optimize any sensor with a fixed footprint and a fixed target geometry. The very idea is to automatically modify the shape of the RX coils in such a manner that the linearity error is reduced. An enabling technology for the optimization of the RX coil geometry is the fast virtual prototyping of the sensor.

Different optimization algorithms have been tested in order to determine which of them provides the best trade-off between the linearity error and time consumption. Among them, PSO was tested by describing RX geometry by means of few control points. After optimization, the non-linearity error was below 0.2%FS, but the time required for optimization was prohibitive. If one would like to better describe geometry by adding the necessary flexibility to reduce the error toward zero, the number of variables to optimize would increase to the number of points representing the geometry of the receivers. In this case, optimization is not feasible with PSO or any other global optimization technique because hundreds or thousands of variables need to be optimized.

The best results were obtained by applying a non-linear least-squares (NLS) solver. This approach, available as an off-the-shelf Intel Math Kernel Library routine, provided the best performance in terms of time consumption and non-linearity error if compared to the other algorithms. The non-linear optimizer with a trust region iteratively searches the geometric shape of the RX coils, expressed as the radius vector h, which minimizes the sensor non-linearity. The problem can be formulated as follows:(4)$$\min_{h\in R^{n}}{∥\theta \left(h\right)-\theta^{\mathrm{id}}∥}_{2}^{2},$$
where $\theta \left(h\right)\in R^{p}$ is the measured vector of the *p* angles for each position of the target, and *n* is the number of points that defines the geometry of the receivers, whereas $\theta^{\mathrm{id}}\in R^{p}$ are the ideal angles of (Equation 1).

However, the optimization problem (Equation 4) has a drawback since the minima reached does not’t assure the same amplitude for both induced voltages. For this reason, the problem is reformulated and the optimizer follows ideally induced voltages instead of ideal linearity since the first condition implies the second. Referring to only one receiver, although optimization is performed in parallel for both of them, RX’s shape is determined by the following:(5)$$\min_{h\in R^{n}}{∥U_{\mathrm{rx}}\left(h\right)-U_{\mathrm{rx}}^{\mathrm{id}}∥}_{2}^{2},$$
where $U_{\mathrm{rx}}\left(h\right)\in R^{p}$ is the vector of *p* induced voltages for each *p* position of the target, $U_{\mathrm{rx}}^{\mathrm{id}}$ includes the ideal voltages, and *n* involves the coordinates that define the geometry of the receivers. A Taylor expansion of the function $U_{\mathrm{rx}}\left(h_{k+1}\right)$, at step k+1, can be written as follows:(6)$$U_{\mathrm{rx}}\left(h_{k+1}\right)≊U_{\mathrm{rx}}\left(h_{k}\right)+J_{k}s,$$
where $J_{k}$ denotes the Jacobian matrix of $U_{\mathrm{rx}}\left(h_{k}\right)$ and vector s is the difference between $h_{k+1}$ and $h_{k}$. By plugging (Equation 6) into (Equation 5), the problem can be written as follows:(7)$$\min_{s\in R^{n}}∥J_{k}s+F\left(h_{k}\right){∥}_{2}^{2},$$
where $F\left(h_{k}\right)=U_{\mathrm{rx}}\left(h_{k}\right)-U_{\mathrm{rx}}^{\mathrm{id}}$. Now it is possible to determine the next step by finding the optimal s. A possible solution may be obtained by solving the following normal equation:(8)$$J_{k}^{T}J_{k}s^{*}=-J_{k}^{T}F\left(h_{k}\right),$$
although other methods can be applied for solving sub-problem (Equation 7) within the trust region, as extensively discussed in [29]. The algorithm ends when the minima, or the maximum number of iterations is attained.

## 5. Measurements

A set of measurements has been performed on various geometries of the sensor and a comparison between the physical prototypes, and their virtual counterpart is provided below. The considered designs are reported in Figure 2. For each design, a Gerber file has been produced and a prototype has been fabricated. In these prototypes, the traces have a thickness of 35 μm and a width of 200 μm, while the thickness of the PCB is 1.6 mm.

An in-house measurement system consisting of a set of motorized and manual precision mechanical stages with 11 degrees of freedom has been built in order to center the target and set the air gap to 1 mm (Figure 3). A signal generator (Siglent SDG6022X) is used in order to feed the TX coil at a frequency of 2.083 MHz and amplitude of 3.5 Vpp. A data acquisition board (Picoscope 5000 Series) acquires the received signals from RX coils and from the TX one. Post-processing consists of extracting the envelope of the received signals.

Table 2 provides, for each considered design, a summary of the features and the relative simulated and measured errors that are described in detail below. The linearity error is computed with respect to the least-square straight line curve, which is not required for passing through any specific data point of the sensor’s output characteristic [30,31]. It is important to remark that the discrepancy between the virtual and the real prototype is due to the fact that it is very difficult to align the latter with the target into its nominal position, even by using precision mechanical stages. Indeed, a misalignment of some tens of microns can be enough to change the behavior of the sensor, especially when the linearity error is very small.

The first design (Figure 2a) that has been considered places the two RX coils: one in the top of the PCB and the other in the bottom. The reason behind this choice is to avoid, as much as possible, the presence of vias that is otherwise necessary when the two coils are placed on the same PCB layer. The shape of the RX coils is ideal, i.e., without optimization. As it can be seen from the measurements performed on this prototype (Figure 4b), the linearity error is higher than 3.5%FS. This error is mainly due to the fact that the amplitudes of the induced voltages are very different between each other because the receiver placed at the bottom of the PCB exhibits a lower signal than the other one (the red line in Figure 4a), because it is further away with respect to the target.

A method to compensate the mismatch between the induced voltages in the two RX coils is to design the receivers on the same layer in order to reduce, as much as possible, the part of the coil that is located in the bottom layer. The second design (Figure 2b) is implemented in this manner, and the effect of this adjustment is a general improvement in the simulated and measured linearity, if compared to Design#1. The measured linearity error is below 0.63%FS (Figure 4d), whereas the amplitudes of RX coil voltages (Figure 4c) tend to reach ideal shapes.

Although the improvement in the linearity error between Design#1 and Design#2 is evident, Design#3 is an attempt to further reduce this error by adding *dummy exits* (Figure 2c). Indeed, linearity should improve by making the sensor more symmetric, since the flux linkage should be theoretically balanced. However, Figure 4f shows that there is not a substantial improvement in linearity even though the physical measurement is slightly more tolerant to misalignments, and it is easier to center the target within an acceptable tolerance.

To achieve a better performance in terms of linearity error, the design optimization procedure described in the former sections was adopted. Design#4 and Design#5 (Figure 2d,e, respectively) are results of the NLS optimization algorithm. Design#4 is optimized starting from Design#1 and by adding the dummy exit, whereas Design#5 has a different arrangement of the RX coils, which is called *interleaved*. The advantage of interleaving is that the RX coils have the same distribution between the top and the bottom of the PCB. The symmetry offered by interleaving together with the dummy exits provides a robust performance to the sensor with respect to the tilt and centering of the target, if compared to all other designs.

Figure 5b,d show that the simulated error improves noticeably if the NLS algorithm is used. Indeed, simulated errors below 0.1%FS were achieved for Design#5 and below 0.01%FS for the interleaved design. The entire optimization of the sensor requires less than half an hour. Nevertheless, the measurements show errors that are higher than the simulated ones. This is due to the measurement setup where the misalignment of the target and the noise of the ADC are responsible for the deviation from the simulated design. Furthermore, this effect becomes more evident when the linearity error is reduced, as the results of Design#4 and Design#5 show. However, the pattern of the virtual prototype is maintained, and this can be observed at the zero crossing where the linearity error is zero.

## 6. Conclusions

In this work, a fast and efficient tool that exploits the surface integral method was used for the virtual prototyping of an absolute ironless inductive position sensor. The tool has been used to automatically optimize the sensor in terms of the linearity error and to reduce the amplitude mismatch between the voltages induced on the two receiving coils. Distinct from other optimization techniques proposed in the literature, the footprint and target geometry of the sensor have been kept fixed, as it happens in most practical cases where the target has to fit some predefined space. The novel technique proposed to reduce the sensor linearity error is based on changing the shape of the RX coils. This is realized with a non-linear least-squares solver that is able to reduce the simulated error toward zero. Indeed, the measurements performed on various prototypes confirm the effectiveness of the simulation and optimization tool by providing results below 0.1%FS without any signal calibration or post-processing manipulation. We stress that the improvements achieved on the virtual prototype are directly reflected in the physical one, despite the differences between their linearity errors that are mainly due to the fact that it is very hard to perfectly center the target in its nominal position. In this respect, designing a sensor that is more tolerant to misalignments will be the objective of further studies.

## 7. Patents

Preliminary results about the optimization method can be found in patent deposited from Gentjan Qama, Mauro Passarotto, and Ruben Specogna with title *Sensor Coil optimization*, number WO2019089095A1, 2019.

## Figures and Tables

**Figure 1 sensors-22-04683-f001:**
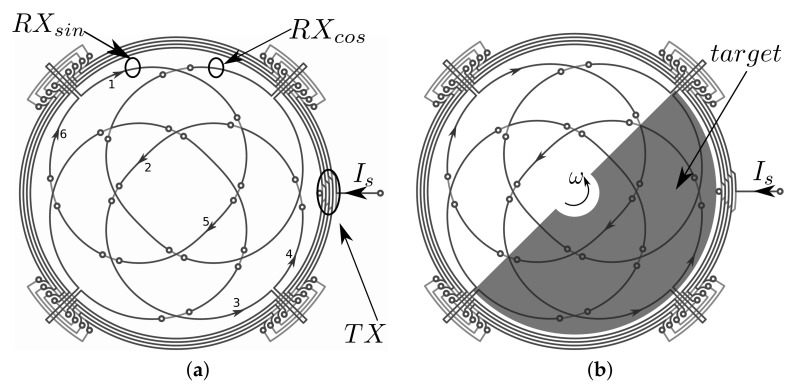
The geometry of a rotary IPS. (**a**) the sensor without the conductive target; the transmitter (TX) is driven by a known alternating current, whereas receivers $RX_{sin}$ and $RX_{cos}$ pick up the induced voltages. (**b**) the sensor with the conductive target covering a portion of the sensor and rotating with angular speed ω; eddy currents flowing in the target perturb the system by means of their reaction magnetic field that superpose to the source field generated by TX coil.

**Figure 2 sensors-22-04683-f002:**
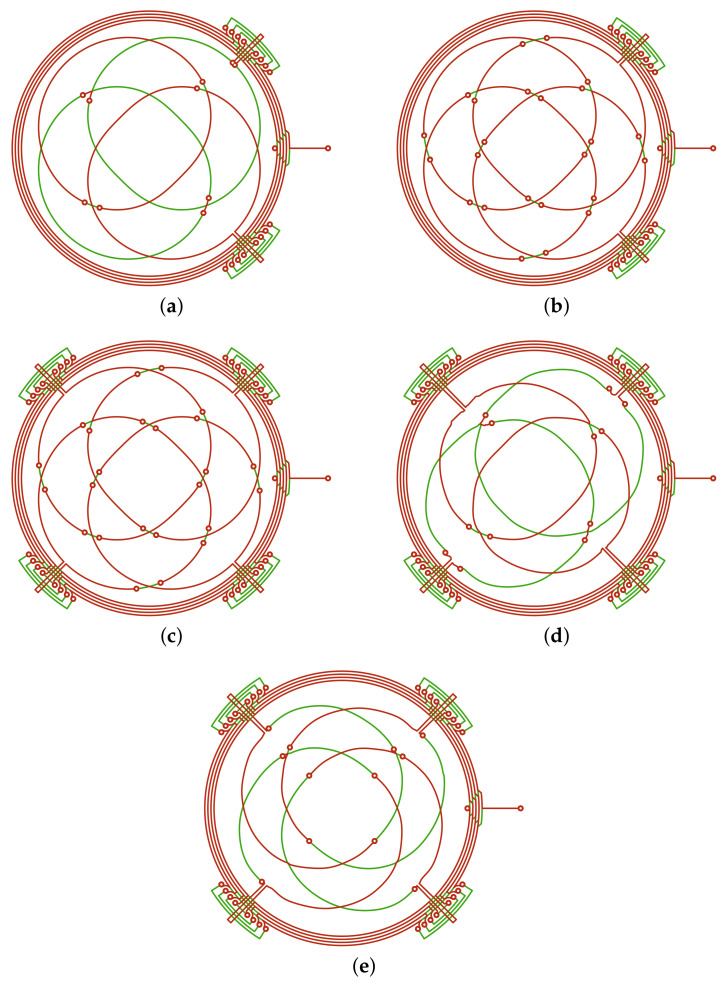
The set of geometries simulated and measured. (**a**) Design#1: receivers placed into separated layers, one on the top and one on the bottom, without optimization; (**b**) Design#2: receivers sharing the top PCB layers, without optimization; (**c**) Design#3: receivers as in (**b**), but with the presence of dummy exits that provide symmetry to the sensor that is missing in the previous designs; (**d**) Design#4: receivers as in (**a**) with dummy exits, optimized with NLS; (**e**) Design#5: receivers connected with interleaved topology and optimized with NLS.

**Figure 3 sensors-22-04683-f003:**
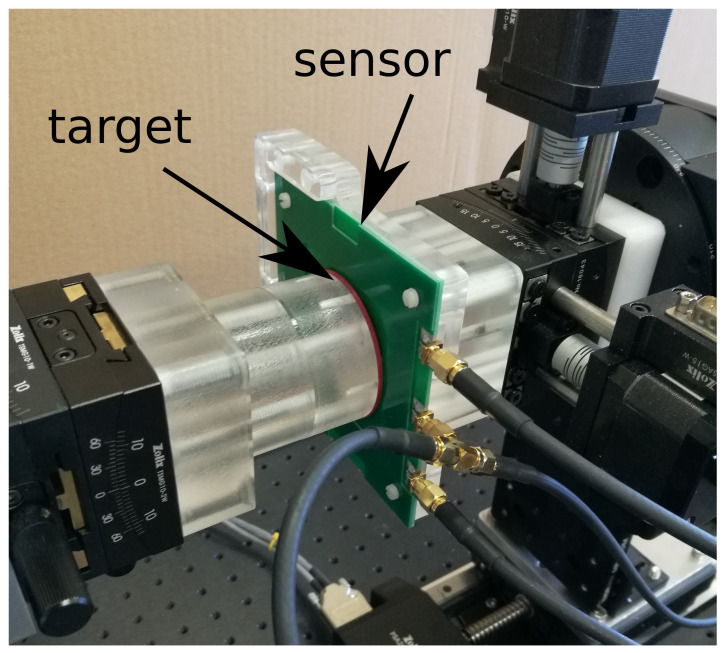
Setup of the measurement for the rotary inductive position sensor. After the centering of the conductive target, the air gap is set to 1 mm.

**Figure 4 sensors-22-04683-f004:**
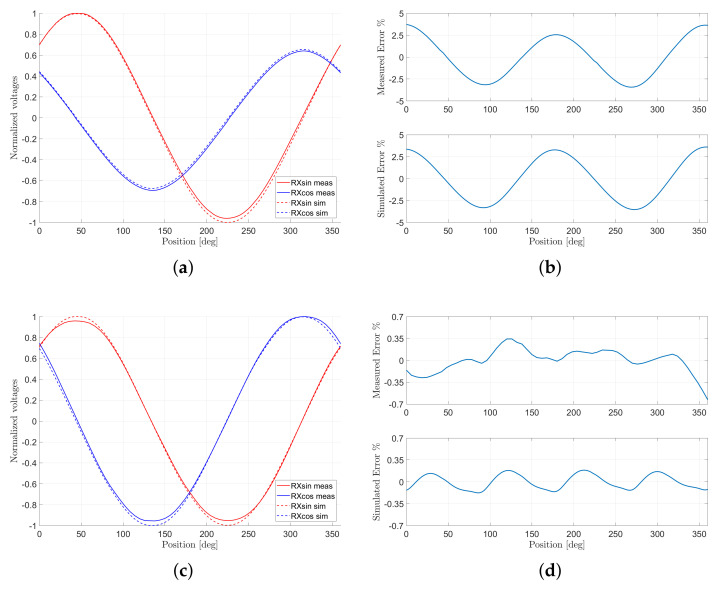
Comparison between the simulation and the measurements for the non-optimized designs. Each raw (from top to bottom) corresponds to Design#1, Design#2 and Design#3, respectively. (**a**,**c**,**e**) show the simulated and measured induced voltages. The displayed values are normalized with respect to the amplitudes of the simulated and measured $U_{\mathrm{rxsin}}$, respectively. (**b**,**d**,**f**) show the linearity errors. The position, which is expressed in degree, refers to electrical angles.

**Figure 5 sensors-22-04683-f005:**
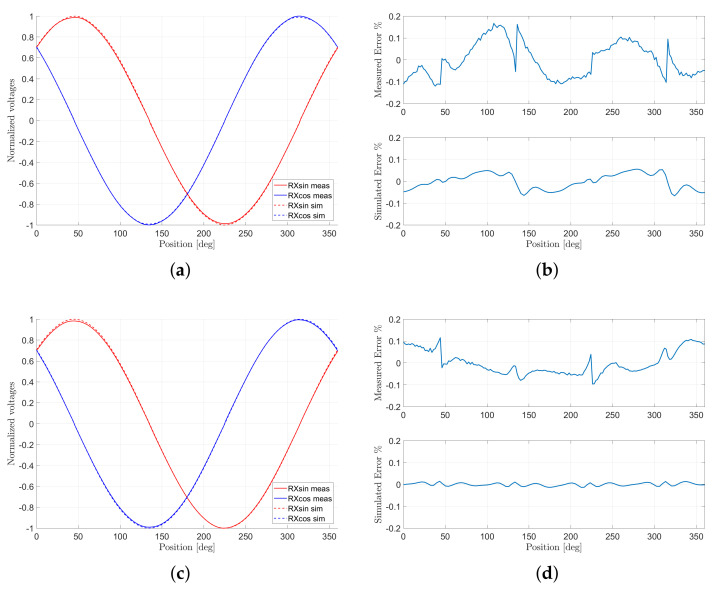
Comparison between the simulation and the measurements for the optimized designs with NLS. Each raw (from top to bottom) corresponds to Design#4 and Design#5, respectively. (**a**,**c**) show the simulated and measured induced voltages. The displayed values are normalized with respect to the amplitudes of the simulated and measured $U_{\mathrm{rxsin}}$, respectively. (**b**,**d**) show the linearity errors. The position, which is expressed in degree, refers to electrical angles.

**Table 1 sensors-22-04683-t001:** Conductivities and relative skin depth at automotive application frequencies for different conductors.

Material	σ (S/m) at 20 °C	δ (μm) at 2.2 MHz	δ (μm) at 5.6 MHz
copper	5.95 ×$10^{7}$	44	28
aluminum	3.77 × $10^{7}$	55	35

**Table 2 sensors-22-04683-t002:** Features and relative simulated and measured linearity error for each tested design.

Design	#1	#2	#3	#4	#5
RX on top and bottom	✓	✗	✗	✓	✓
RX only on top	✗	✓	✓	✗	✗
RX with dummy exit	✗	✗	✓	✓	✓
RX optimized with NLS	✗	✗	✗	✓	✓
RX interleaved	✗	✗	✗	✗	✓
Simulated error (%FS)	3.62	0.19	0.25	0.07	0.01
Measured error (%FS)	3.74	0.63	0.44	0.17	0.11

## Data Availability

Not applicable.

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
