# Peer review of "Design Optimization of PCB-Based Rotary-Inductive Position Sensors"

_sensors, 2022, doi:10.3390/s22134683_

Round 1

Reviewer 1 Report

I have no comments on the article under consideration. I appreciate its undeniable originality, good logical structure and especially the balanced ratio between modeling and experimental results. The results obtained by optimizing the shape of the RT windings are convincing and remarkable, especially in that these excellent properties are achieved without calibration of the sensors.
Overall, the article is beneficial, I wish the authors a lot of success in their future work.

Author Response

We would like to thank the reviewer for taking the time to review our manuscript. We are also glad that he enjoyed reading the article. It's for sure a positive stimulus for future research and collaborations in these topics.

Sincerely,

Aldi Hoxha

Reviewer 2 Report

Authors showed new design optimization method of radiometric rotary IPS on the PCB. This proposed method could reduce linearity error and amplitude mismatch of the receiving coils.

English grammar looks fine. Simulated and measured results looks good and have good matched. Thus, the manuscript could be minor revision after addressing following comments.

1.Please correct Fig. to Figure in entire manuscript according to MDPI styles.

2.Please do not use capital letters for RSM and PSO after line 199.

3.Authors had better mention important measured results in conclusion section

4.Please use abbreviated journal names in the reference section.

5.Please provide the city and country for book in the reference section.

6.Please provide ref. (Each of those has~) with ref.(https://www.hindawi.com/journals/jhe/2017/6580217/).

7.It looks like 2 references in Ref. [31] so correct that.

8.In Fig.4, authors showed measured results for +1/-1 voltage ranges. Are there any effects if higher voltages or higher frequencies are applied.

9. Authors had better mention future work in conclusion section.

10. In Figure 3, authors used some equipment. Please show the spec. of the used equipment for measured results.

11. Figure 5 fonts looks a little bit small to be seen.

12. In Figure 1, there are RXsin and RXcos and TX. I am wondering whether TX has fixed location. 

Author Response

We would like to thank the reviewer for his constructive comments and suggestions which have improved different parts of the article.

Response to the reviewer:

1.Please correct Fig. to Figure in entire manuscript according to MDPI styles.
We apologize for our error. We corrected this point.

2.Please do not use capital letters for RSM and PSO after line 199.
We modified this point from "Response Surface Method" to "response surface method" and the same was done with the particle swarm optimization.

3.Authors had better mention important measured results in conclusion section
We added the important point that the measured linearity error have been obtained without calibration or post-processing manipulation.

4.Please use abbreviated journal names in the reference section.
When possible, the journals' names have been abbreviated.

5.Please provide the city and country for book in the reference section.
We added the city and country of the books.

6.Please provide ref. (Each of those has~) with ref.(https://www.hindawi.com/journals/jhe/2017/6580217/).
We don't understand this comment. What should we do?

7.It looks like 2 references in Ref. [31] so correct that.
The reference has been corrected.

8.In Fig.4, authors showed measured results for +1/-1 voltage ranges. Are there any effects if higher voltages or higher frequencies are applied.
The purpose of those figures is to illustrates the discrepancy between the measured and the simulated voltages. This values are normalized the wrt the 
induced voltage on the sine coil. 
If higher voltages are applied to the transmitting coil there would be a benefit in terms of the amplitude of the received signal whereas the frequency
is dictated from the regulations used in automotive applications.

9. Authors had better mention future work in conclusion section.
We mentioned already future work and research topics such as the study and the design of inductive position sensors 
which can be more robust whenever misalignements occur.

10. In Figure 3, authors used some equipment. Please show the spec. of the used equipment for measured results.

The signal generator Siglent SDG6022X was used for the excitation of the transmitting coil and the oscilloscope Picoscope 5000 Series
for the acquisition of the signals. Below the references about the instrumentation used for the measurements:

https://www.siglent.eu/product/1138498/siglent-sdg6022x-200mhz-function-arbitrary-waveform-generator

11. Figure 5 fonts looks a little bit small to be seen.
Thank you for the suggestion. We increased the width of the images.

12. In Figure 1, there are RXsin and RXcos and TX. I am wondering whether TX has fixed location
The radius of the transmitter, which define the maximum dimension of the sensor, is fixed. The only modifiable parameters are the points 
representing the receivers which can move within a certain bound.